# The Significance of Phenotyping and Quantification of Plasma Extracellular Vesicles Levels Using High-Sensitivity Flow Cytometry during COVID-19 Treatment

**DOI:** 10.3390/v13050767

**Published:** 2021-04-27

**Authors:** Igor Kudryavtsev, Olga Kalinina, Vadim Bezrukikh, Olesya Melnik, Alexey Golovkin

**Affiliations:** 1Almazov National Medical Research Centre, 197341 St. Petersburg, Russia; igorek1981@yandex.ru (I.K.); olgakalinina@mail.ru (O.K.); ikar19882012@gmail.com (V.B.); orangelove@yandex.ru (O.M.); 2Institute of Experimental Medicine, 197376 St. Petersburg, Russia

**Keywords:** COVID-19, SARS-CoV-2, plasma extracellular vesicles, exosomes, high-sensitivity flow cytometry, phenotyping extracellular vesicles

## Abstract

New investigation results point to the potential participation of extracellular vesicles (EVs) in the pathogenesis of coronavirus infection, its progression, and mechanisms of the therapy effectiveness. This dictates the necessity to transfer scientific testing technologies to medical practice. Here, we demonstrated the method of phenotyping and quantitative analysis of plasma EVs based on differential centrifugation, immunostaining, and high-sensitivity multicolor flow cytometry. We used EV markers that were potentially associated with SARS-CoV-2 dissemination via vesicles and cell-origination markers, characterizing objects from different cell types that could influence clinical manifestation of COVID-19. Plasma levels of CD235a+ and CD14+ EVs in patients with moderate infection were significantly increased while CD8+ and CD19+ EVs were decreased comparing with HD. Patients with severe infection had lower levels of CD4+, CD19+, and CD146+ EVs than HD. These findings demonstrate that EV concentrations in COVID-19 are severity related. Moreover, the three-point dynamic assessment demonstrated significant loss of CD63+ and CD147+ plasma EVs. The used method can be a convenient tool for vital infection pathogenesis investigation and for COVID-19 diagnostics.

## 1. Introduction

Coronavirus disease 2019 (COVID-19), a disease caused by severe acute respiratory syndrome coronavirus 2 (SARS-CoV-2), first emerged in Wuhan, China, in December 2019, and was declared a pandemic by the World Health Organization (WHO) in March 2020. In critical cases, the infection can cause acute respiratory distress syndrome (ARDS), cytokine storm, organs failure, septic shock, and blood clots [1,2].

Currently, many references show the potential participation of extracellular vesicles (EVs) in the pathogenesis of coronavirus infections, their progression, or, conversely, in the mechanisms of therapy effectiveness [3,4]. Virus-infected cells release EVs that are implicated in infection through transferring viral components [1]. EVs may contribute to spreading the virus by using the same cell receptors for transfer [1]. Moreover, EVs secreted by different cell subsets could participate in pathogenesis by influencing coagulation, inflammation, microcirculation, etc.

Applying investigative results of quantitative and qualitative concentrations of plasma extracellular vesicles is relevant in clinical applications, including when monitoring the treatment of a new coronavirus infection. While utilizing scientific approaches in clinics, it is necessary to choose utilitarian research methods based on fairly simple EV isolation techniques, study their quantitative and qualitative compositions, as well as phenotyping investigating their potential participation in COVID-19 pathogenesis, progression, and/or complication development. 

Based on these approaches, differential centrifugations with adequate controls of residual cell elimination seem to be applicable methods of EV isolation [5,6]. A new generation of high-sensitivity flow cytometers (hs-FCM) have improved the ability of investigating biological objects smaller than 200 nm, because of multicolor monoclonal antibodies specific to EVs and cell receptors, given the opportunity to investigate EVs quantitatively and qualitatively [5,7,8,9].

The aim of the study was to demonstrate the applicability of a routine isolation and phenotyping method of plasma extracellular vesicles using multicolor hs-FCM in patients with COVID-19, and to demonstrate the potential diagnostic and pathogenesis impact of the performed method.

## 2. Materials and Methods

The study was performed with 21 COVID-19 patients with moderate infections (9 patients, 62 (50;65) years old) or severe (12 patients, 60 (52;79) years), undergoing treatment in the in-patient departments at Almazov National Medical Research Center (St. Petersburg, Russia), and 10 apparently healthy donors (HD) (36 (32;49) years old). The research was approved by the local ethics committee (protocol no. 2209-20; 21 September 2020), and complied with the Helsinki Declaration. All patients included in the protocol provided informed consent. 

COVID-19 diagnoses were based on epidemiological and clinical data, detection of SARS-CoV-2 RNA using a “SARS-CoV-2/SARS-CoV” PCR detection kit (DNA-technology TC, Russia), and anti-SARS-CoV-2 IgM/IgG in the patients’ plasma using a “DS-ELISA-anti-SARS-CoV-2″ detection kit (RPC «Diagnostic Systems», Nizhny Novgorod, Russia).

According to the clinical data and severity of the respiratory insufficiency, all patients were divided into two groups: moderate (SpO2 < 95%, respiratory rate > 22/min, qSOFA < 2) and severe (SpO2 < 93%, respiratory rate > 30/min, PaO2/FiO2 ≤ 300 mm Hg, qSOFA > 2) COVID-19. Both groups had significantly (*p* < 0.01) lower levels of hematocrit, lymphocytes, eosinophils, and basophils compared with HD. Moreover, patients with severe infection had decreased white blood cells and monocyte rates comparing with healthy donors, but not of erythrocytes and platelets. Patients with severe infections had significantly (*p* < 0.01) higher lung damage, 60 (48;68)% based on the CT (computed tomography) results than with moderate (40 (32;52)%) COVID-19.

All patients received oxygen therapy, depending on their conditions, ranging from 1 to 21 days. Moreover, 49% of patients received short courses of prednisolone or dexamethasone therapy for up to 4 days. Patients from both groups, during the treatments, had positive clinical dynamics, and they were safely discharged from the hospital.

Peripheral blood samples were collected into vacuum test tubes containing K_3_-EDTA and then processed to prepare the plasma samples for extracellular vesicles analysis. Samples were collected before the beginning of the in-patient department treatment (1st point), after 6–9 days (2nd point), and additionally 6–9 days after the previous examination (3rd point). Clinical analysis was performed by using Cell-DYN Ruby-Hematology Analyzer (Abbot, Chicago, IL, USA).

We performed a previously described and experimentally approved method of extracellular vesicles isolation from the whole blood by differential centrifugation [10]. Briefly, whole blood samples were centrifuged at 1500 g for 10 min, followed by transfer of plasma to a new tube that was centrifuged twice—first at 1500 g for 10 min and then the obtained supernatant was centrifuged at 3000 g for 10 min. 

Obtained plasma was engaged in immunofluorescence of the EV staining process. Phenotyping of plasma EVs was carried out using the following monoclonal antibodies: CD235a PE (BioLegend, San Diego, CA, USA), CD41 AF488 (BioLegend, San Diego, CA, USA), CD45 KrO (Beckman Coulter, West Sacramento, CA, USA), CD 3 AlexaFluor750 (Beckman Coulter, West Sacramento, CA, USA), CD4 PE (Beckman Coulter, West Sacramento, CA, USA), CD8 PC5.5 (Beckman Coulter, West Sacramento, CA, USA), CD19 FITC (BioLegend, San Diego, CA, USA), CD14 APC (Beckman Coulter, West Sacramento, CA, USA), CD146 PE (Beckman Coulter, West Sacramento, CA, USA), CD31 FITC (BioLegend, San Diego, CA, USA), CD34 PE/Dazzle (BioLegend, San Diego, CA, USA), CD90 PE/Cy7 (BioLegend, West Sacramento, CA, USA), CD105 APC (Beckman Coulter, West Sacramento, CA, USA), CD9 PCy7 (BioLegend, San Diego, CA, USA), CD63 APC (BioLegend, San Diego, CA, USA), CD147 PE (BioLegend, San Diego, CA, USA). A total of 100 µL of plasma was stained with 1 µL of appropriate antibodies, for 20 min in the dark at room temperature.

Analysis of EVs was performed using CytoFLEX S (Beckman Coulter, West Sacramento, CA, USA) flow cytometer. Instrument calibration setup was performed using the Cytometry Sub-Micron Particle Size Reference Kit, Molecular probes by Life Technologies), and Megamix-Plus FSC and Megamix-Plus SSC (Biocytex, Marseille, France) containing FITC-labeled reference beads of various diameters. All control samples, detergent effectiveness treatments, and possible false-positive or false-negative controls, were performed in accordance with the previously published requirements [7,11] and as previously described by our group [10,12]. Stained sample dilution was performed in accordance of prevention of coincidence effects, as previously described [10,12,13]. Detergent sensitivity of EVs is presented in the Appendix A.

Statistical analysis was performed using Statistica 7.0 (StatSoft, Oklahoma, OK, USA) and GraphPad Prism 8 (GraphPad software Inc., San Diego, CA, USA) software. All results were presented as median and interquartile range—Me (25;75). The dispersion analysis was performed using ANOVA statistics. The differences between groups were analyzed using the nonparametric Man–Whitney U-test. The differences between groups during the treatment were analyzed using the Kruskal–Wallis test. Significances were set at *p* < 0.01.

## 3. Results

Some significant differences were observed between plasma levels of EVs in HD and patients with moderate and severe COVID-19 before the beginning of the in-patient department treatment. In particular, plasma levels of CD235a+ and CD14+ EVs in patients with moderate infections significantly increased. CD8+ and CD19+ EVs decreased compared with HD (Figure 1). Patients with severe infections demonstrated lower levels of CD4+, CD19+, and CD146+ EVs than HD. It was noteworthy that the higher severity of the infection led to significant decreases of CD235a+, CD45+, CD19+, and CD14+ EVs levels.

The three-point dynamic assessment demonstrated significant progressive loss of CD63+ and CD147+ plasma EVs (Figure 2). Noteworthy, plasma levels of CD63+ EVs were lower at each treatment time-point in patients than in HD (*p* = 0.001).

The treatment-related dynamics of only CD235a+, CD4+, CD8+, CD19+, and CD146+ EVs were significant (Figure 2).

## 4. Discussion

All types of extracellular vesicles potentially participating in COVID-19 progression could be divided in two groups: 1) related to several glycoproteins and/or receptors that are essential for a complete fusion event [14], and 2) able to take part in the pathogenesis process, particularly in immune response, endothelial damage, ARDS, microcirculatory disorders, or cytokine storming [15].

Thus, the first group of EVs could be associated with tetraspanins (CD9, CD63, CD81, CD82 molecules), angiotensin-converting enzyme 2, CD147, etc. Several tetraspanins that are enriched in exosomal membranes [16] may also participate in coronavirus fusion events [17]. In particular, CD9 molecules play a key role in loading exosomal cargo by the protein–protein interaction network [18], supporting a pivotal role in loading COVID-19 virus proteins [1,19,20], as well as participating in exosome biogenesis [18]. Gunasekaran et al. demonstrated that circulating exosomes isolated from patients with respiratory viral infections contained specific viral antigens of respiratory syncytial virus, coronavirus, and rhinovirus, respectively [21]. Moreover, some findings show SARS-CoV-2 within the vacuoles or double membrane vesicles within the host cells [22,23], suggesting that there is a possibility for the use of the exosomal cellular transport as a mode of systemic SARS-CoV-2 viral dissemination and COVID-19 reactivation [4]. Membrane vesicles containing viral particles released from infected cells circulate systemically and disseminate to reach distant tissues [24] and organs, including the vasculature system [25]. Other tetraspanins, such as CD63, CD81, and CD82 released from viral infected cells could also participate in viral component transfer for EBV [26], HCV [27], HIV [28,29], or other viruses [5].

ACE2 (angiotensin-converting enzyme 2) is considered the main target receptor for SARS-CoV-2 virus infection, expressed in alveolar epithelial type II cells, intestinal epithelial, bronchial epithelial cells, vascular endothelial cells, heart, etc. [30,31]. Recent experimental studies demonstrated that CD147, a type II transmembrane glycoprotein that belongs to the immunoglobulin superfamily, might also act as a receptor mediating SARS-CoV-2 entry through binding to the S protein [30,32]. CD147 is a widely expressed protein in many cell types, including hematopoietic, epithelial, and endothelial [32], thereby, emphasizing the broad possibilities of its involvement in the infection process.

None of the investigated CD63+, CD9+, or CD147+ EVs were at a higher levels than in HD; moreover, CD63+ EVs level were even low. During the treatment, CD63+ and CD147+ EVs demonstrated reliable dynamics to decrease their levels. These results may be in concordance with theories of involvement of EVs, particularly containing tetraspanins in the viral transport and infection progression. Thus, decreases of CD63+ and CD147+ EVs levels demonstrate loss of COVID-19 infection progression, and become diagnostic markers.

The second group of plasma vesicles is presented with EVs of different cell origins, capable of participating in pathogenesis and manifestation of the disease. Platelets are one of the main sources of EVs in the plasma. Produced vesicles could demonstrate both pro-inflammatory and anti-inflammatory effects via interactions with endothelial cells and leukocytes [5]. Their pro-inflammatory potentials are related with the ability to carry cytokines, such as interleukin 1 beta (IL-1b), IL-6, and tumor necrosis factor alpha (TNFα), which stimulate neutrophil adhesion to endothelium and transendothelial migration [33,34]. However, platelet EVs were shown to exert anti-inflammatory effects, as they decrease c-c-motif chemokine ligand 4, TNFα, and colony-stimulating factor-1 release from macrophages [35], reduce TNFα, and IL-8 secretion, and inhibit IL-17 and interferon γ production by regulatory T-cells [5,36].

Red blood cells (RBC) are also a significant source of plasma EVs and their levels increase in systemic inflammation [5,37]. RBC EVs can cause oxidative damage to phospholipids on cell membranes [38] and were also implicated in neutrophil priming and CD11b upregulation for enhanced adhesiveness to vascular endothelium [5].

All subsets of white blood cells can produce EVs, with the neutrophil being the main source, especially during acute inflammation [5]. Monocyte EVs are known to induce endothelial inflammation via the nuclear factor κB (NF-κB) pathway [39], or by transfer cargo carrying IL-1β, and components of the inflammasome pathway [39]. Moreover, monocyte EVs could increase nutritive stress [39] and promote thrombogenicity [40]. Lymphocyte EVs are relatively less well studied, but it is known that T-cell derived EVs play a crucial role in the chronic stage of inflammation or autoimmune disease. Their effects could be realized via decreased nitric oxide (NO) production and increased ROS (reactive oxygen species) production in endothelial cells [5]. However, there is almost no information about phenotyping and concentrations of white blood cell plasma vesicles in patients and healthy donors, despite their potential impact on pathogenesis of different pathological processes.

Endothelial-derived EV production can be induced by TNFα, IL-1β, thrombin, C-reactive protein, plasminogen activator-1, high glucose conditions, and hypoxia [41]. Endothelial cell derived EVs can directly target vascular endothelium or bind leukocytes and contribute to their activities in inflammatory disorders [5,41].

Levels of red blood cell-derived CD235a+ and leukocyte-derived CD45+, CD19+, and CD14+ EVs demonstrated severity-dependent level reduction (Figure 1). Moreover, the most numerous CD235a+ and platelet-derived CD41+ EVs were elevated in patients comparing to HD. Significant changes in levels of plasma EVs in patients with different severity of infections demonstrated participation of red blood cell- and leukocyte-derived vesicles in COVID-19 pathogenesis, which cannot be declared about most investigated endothelial-derived EVs, except CD146+.

Whilst treating, the decrease of CD235a+, CD4+, CD8+, and CD19+ EVs demonstrated their values in pathogenesis of the infection or ability to additionally concretize treatment effectiveness. Additional attention can be placed on the dynamics of plasma CD19+ EVs because of statements surrounding the diagnostic and prognostic effectiveness of this marker in some tumors [42]; our results demonstrate a progressive decrease during the effective treatment. 

Interestingly, members of the coronavirus family induce typical membrane vesicles, with an average diameter of 300 nm [4,43,44]. Formation of larger packets (1–5 µm) can occur after fusion of the outer membranes of some CoV-induced vesicles [43]. Thus, being infected with coronavirus cells can produce virus-containing vesicles expressing molecules typical for the cells, with the size detectable by hs-FCM [7,8,9]. Hs-FCM has no ability to measure the size of the objects, but gives the opportunity to compare light scatter characteristics of identified EV populations, with polystyrene beads, with a known diameter (Figure 3). The results of side scatter intensity distribution from different EVs demonstrate their heterogeneity, probably related to their different sizes. Moreover, most CD63+, CD9+, and CD147+ EVs show light scatter characteristics comparable to polystyrene beads up to 500 nm in diameter, which is consistent with the above hypothesis on virus-containing vesicles.

Thus, our understanding of influence on pathogenesis or clinical significance of plasma levels of different cell type-derived extracellular vesicles, especially in COVID-19, is at an insufficient level, but using the described routine methods could significantly increase our knowledge in this.

## 5. Conclusions

Differences in concentrations of some EVs in COVID-19 are severity-related. Plasma levels of CD235a+ and CD14+ EVs in patients with moderate infections significantly increased, while CD8+ and CD19+ EVs decreased compared with HD. Patients with severe infections had lower levels of CD4+, CD19+, and CD146+ EVs than HD.

The three-point dynamic assessment demonstrated significant loss of plasma CD63+ and CD147+ EVs. 

Phenotyping and quantification of plasma EVs, using centrifugation aimed at eliminating residual cells, and high-sensitivity multicolor flow cytometry, could be convenient tools for COVID-19 diagnostics.

## Figures and Tables

**Figure 1 viruses-13-00767-f001:**
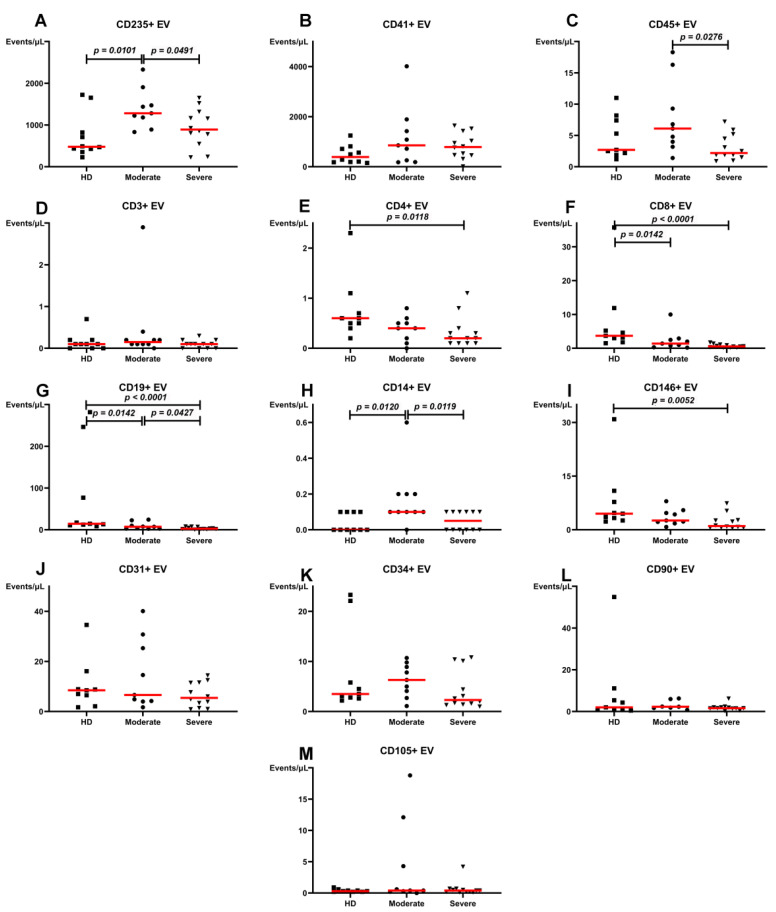
Concentration of plasma extracellular vesicles (EVs) in healthy donors, patients with moderate or severe COVID-19 at the beginning of in-patient department treatment, events/µL. (**A**). CD235a+ EV. (**B**). CD41+ EV. (**C**). CD45+ EV. (**D**). CD3+ EV. (**E**). CD4+ EV. (**F**). CD8+ EV. (**G**). CD19+ EV. (**H**). CD14+ EV. (**I**). CD146+ EV. (**J**). CD31+ EV. (**K**). CD34+ EV. (**L**). CD90+ EV. (**M**). CD105+ EV. Red lines represent median values.

**Figure 2 viruses-13-00767-f002:**
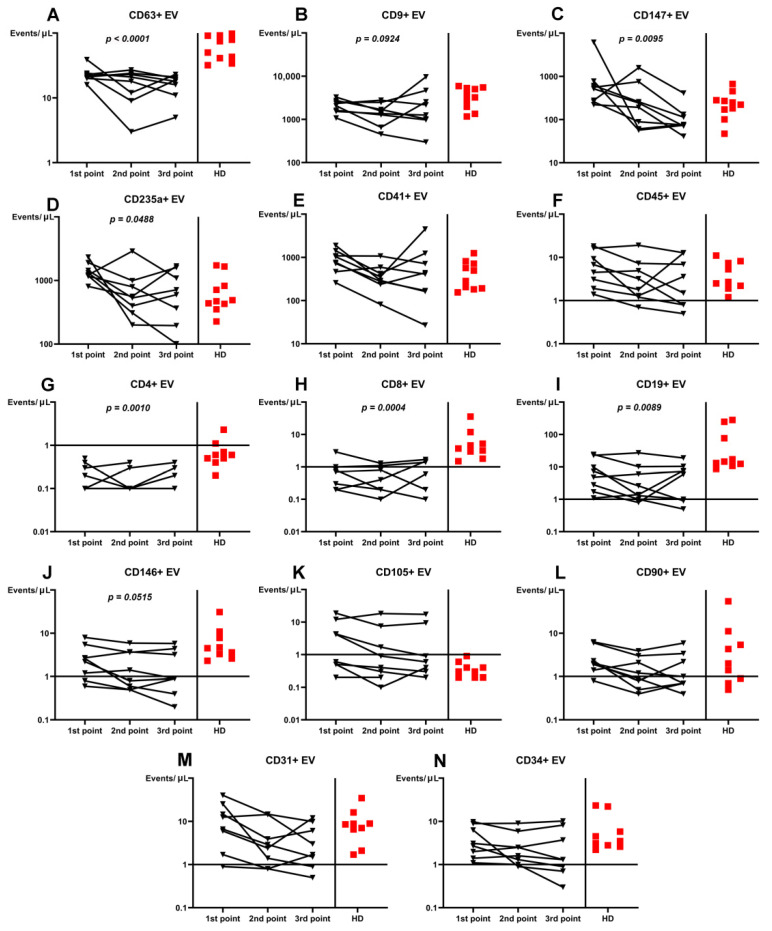
Dynamics in plasma concentration of extracellular vesicles (EVs) in COVID-19 patients during the treatment compared with healthy donors (HD). (**A**). CD63+ EV. (**B**). CD9+ EV. (**C**). CD147+ EV. (**D**). CD235a+ EV. (**E**). CD41+ EV. (**F**). CD45+ EV. (**G**). CD4+ EV. (**H**). CD8+ EV. (**I**). CD19+ EV. (**J**). CD146+ EV. (**K**). CD105+ EV. (**L**). CD90+ EV. (**M**). CD31+ EV. (**N**). CD34+ EV. Results represented in log10 scale.

**Figure 3 viruses-13-00767-f003:**
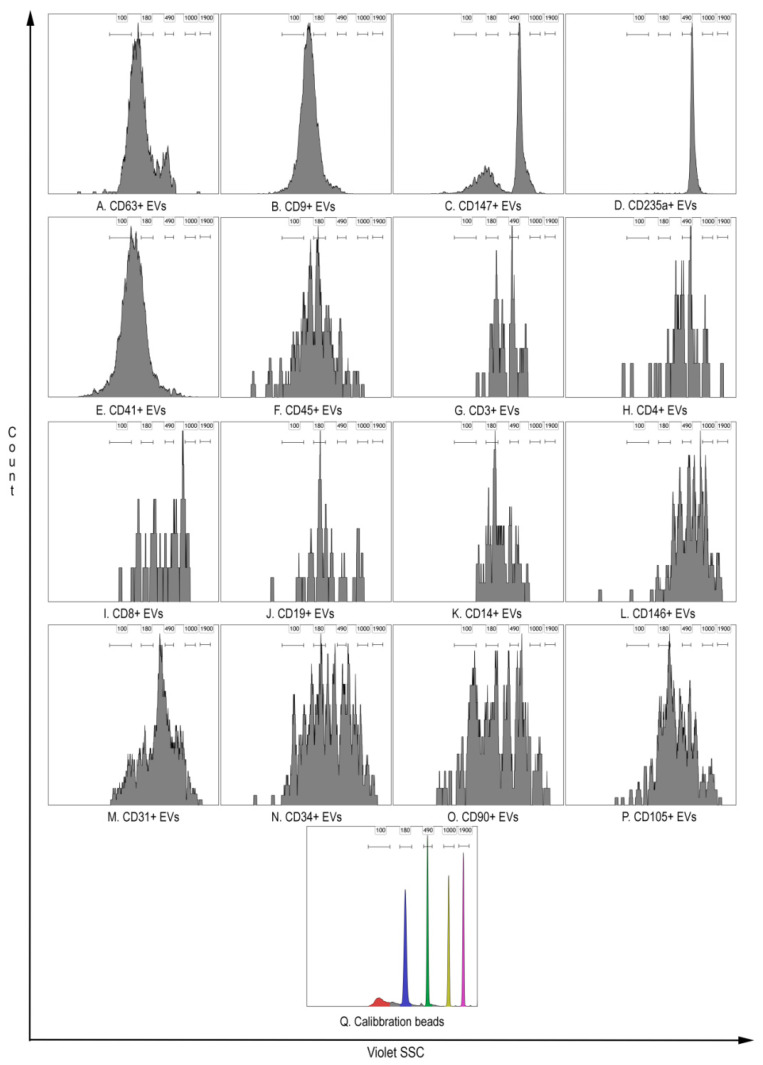
Representative results of side scatter intensity distribution detected from 405 nm violet laser characterizing EVs heterogeneity compared with polystyrene beads with known diameters. (**A**). CD9+ EV. (**B**). CD9+ EV. (**C**). CD147+ EV. (**D**). CD235a+ EV. (**E**). CD41+ EV. (**F**). CD45+ EV. (**G**). CD3+ EV. (**H**). CD4+ EV. (**I**). CD8+ EV. (**J**). CD19+ EV. (**K**). CD14+ EV. (**L**). CD146+ EV. (**M**). CD31+ EV. (**N**). CD34+ EV. (**O**). CD90+ EV. (**P**). CD105+ EV. (**Q**). Sub-micron Particle Size Reference Kit Beads 100, 190, 490, 1000, and 1900 nm diameter.

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
