# Peer review of "The Significance of Phenotyping and Quantification of Plasma Extracellular Vesicles Levels Using High-Sensitivity Flow Cytometry during COVID-19 Treatment"

_viruses, 2021, doi:10.3390/v13050767_

Round 1
Reviewer 1 Report
The manuscript by Kudryavtsev et al is devoted to the phenotyping and quantification of plasma extracellular vesicles (EV) sub-populations in COVID-19 patients. Currently, there are several publications on the possible role of EV in the pathogenesis of SARS-CoV-2 infection, so the high significance of the problem is beyond doubt.
Numerous questions are raised by the design of the study, experimental results and their interpretation.
According to the "Materials and Methods" section the EVs fraction isolation was made using several steps of centrifugation at maximum 3000 g. As far as I know EV isolation is made using ultracentrifugation at 120000g (for example, see Brennan, K., Martin, K., FitzGerald, S.P. et al. A comparison of methods for the isolation and separation of extracellular vesicles from protein and lipid particles in human serum. Sci Rep 10, 1039 (2020). https://doi.org/10.1038/s41598-020-57497-7). For the subsequent staining with antibodies it probably might be sufficient (a highly purified EV fraction is not needed?), but it should be properly explained in the text.
The obtained plasma samples were stained with 16 monoclonal antibodies against a number of CD markers and analyzed using a Cytoflex S flow cytometer (Beckman Coulter, USA). Phrase "multicolor flow cytometry" in the manuscript title do not have a special meaning and rather mislead the reader. Although the labels bound to 16 used monoclonal antibodies had different spectral properties, the samples were stained with antibodies only individually, and the “multicolor” function was not used in any way. In my opinion, the word "multicolor" should be removed from the title.
The choice of a set of CD markers in the study is poorly explained. The authors write the following: «We used EVs markers that are potentially associated with SARS-CoV-2 dissemination via vesicles and cell-origination markers characterizing objects from different cell types that could influence on clinical manifestation of the COVID-19». In my opinion, clarification is needed which CDs from the selected set mark the EV of which types of cells (if so), and how exactly they may be associated with the pathogenesis of COVID-19. If it is difficult to make these clarifications, the choice must be explained in some other way. In the current revision of the manuscript, the 16 CD markers appear to be randomly selected and listed in completely random order.
Fig. 1 gives results on the number of vesicles positive to various CDs, and compares the groups "healthy donors", "moderate COVID-19" and "severe COVID-19". The histograms are shown only for 13 markers out of 16. At the same time, CD3, CD4, CD19, CD90 and CD105 - positive vesicles are registered with a frequency of the order of 1 event per 1 μl of the sample (i.e., essentially not registered), and do not show differences between the groups. In my opinion, these results should not be included in Fig. 1, as they have no meaning.
Figure 1 is entitled «Concentration of extracellular vesicles…. in the debut of the infection». It means that blood samples were taken at the beginning of hospital treatment of patients. However, the moment of hospitalization can in no way be called the debut of infection. Obviously, it was necessarily preceded by a period of home treatment of certain duration. Therefore, it is better to change the name of the picture to "…start of treatment….". And in the characteristics of patients in "Materials and Methods", if possible, indicate information about how much time has passed since infection (or at least since a positive PCR test).
Figures 2 and 3 are exactly the same type and are called the same - «Dynamics in plasma concentration of extracellular vesicles (EVs) in COVID-19 patients during the treatment comparing with healthy donors (HD)». Only CD markers differ - there are 3 of them in Fig. 2 (CD63 +, CD9 +, CD147 +) and 11 others in Fig. 3. It is not clear what the authors were guided by when dividing this data array into 2 figures, and why they divided the data in this way.
In addition, I was unable to find information in the manuscript whether these results on the dynamics of CD markers in extracellular vesicles relate to a severe or moderate course of COVID-19, though this is very important.
Further, the total duration of observation in this case was 12-18 days. It is quite obvious that during this time the condition of some patients should have improved (according to the criteria chosen by the authors), and they should have moved from the “severe course” group to the “moderate course” group, and the condition of some patients possibly should have worsened, with the opposite dynamics. In graphs shown in Fig. 2 and 3, these most important factors are not indicated or commented on.
Regarding to Fig. 3, the authors write “The treatment-related dynamics of only CD235a+, CD4+, CD8+, CD19+, and CD146+ EVs were significant (Figure 3) but among them just CD235a+ and CD19+ are able to be reasonably detectable». If so, then serious questions arise about the significance of all the data in the article in Fig. 1, 2, 3, except for those related to two well-detectable proteins. To clarify the performance and specificity of the antibodies used and to further verify the research results, it would be advisable to demonstrate positive controls and use an alternative research method, for example, Western blotting.
The “Discussion” section is well structered, it demonstrates knowledge and appropriate citation of literature on the problem, but practically does not focus on the authors' own data. It is necessary to analyze and logically link the obtained own results with the existing level of knowledge on the problem, and not be limited to a simple statement of certain changes in the concentrations of CD - markers on the surface of extracellular vesicles.
In lane 158 it is written that “Results of Gunasekaran demonstrated that COVID-19 virus infected cells produce exosomes containing virus particles [21]”. I didn’t find any information in the reference [21] about SARS-CoV-2 virus particles in exosomes. Moreover the manuscript was published online on 2020 Jan 21 and can’t contain any experimental data on SARS-CoV-2. The further statements on EVs, that contain virus particles, are also controversial. EVs from the virus-infected cells are able to contain virus proteins and RNA fragments, some (+)RNA viruses use intracellular EVs maturation pathways for replication and morphogenesis.
In my opinion, the manuscript contains poorly systematized and insufficiently analyzed data and it is necessary to exclude the unsignificant results. The manuscript requires major revision.
Author Response
We appreciate reviewer for the valuable remarks, questions and comments. We are very pleased that our work and our manuscript trigger so many questions. We did all the best to answer the questions. Thus, we hope that performed corrections and clarifications have lead to increased quality of the manuscript.
Prof. Alexey Golovkin and co-authors.
The manuscript by Kudryavtsev et al is devoted to the phenotyping and quantification of plasma extracellular vesicles (EV) sub-populations in COVID-19 patients. Currently, there are several publications on the possible role of EV in the pathogenesis of SARS-CoV-2 infection, so the high significance of the problem is beyond doubt.
Numerous questions are raised by the design of the study, experimental results and their interpretation.
According to the "Materials and Methods" section the EVs fraction isolation was made using several steps of centrifugation at maximum 3000 g. As far as I know EV isolation is made using ultracentrifugation at 120000g (for example, see Brennan, K., Martin, K., FitzGerald, S.P. et al. A comparison of methods for the isolation and separation of extracellular vesicles from protein and lipid particles in human serum. Sci Rep 10, 1039 (2020). https://doi.org/10.1038/s41598-020-57497-7). For the subsequent staining with antibodies it probably might be sufficient (a highly purified EV fraction is not needed?), but it should be properly explained in the text.
Actually ultracentrifugation is the main method for extracellular vesicles isolation. Meanwhile ultracentrifugation at 120000 g is the method for EV fractioning and exosome isolation. However, currently the utilitarian methods of EVs investigations are the most valuable for implementation of scientific approaches into clinics. Our used centrifugation method was aimed to eliminate residual cells from plasma but not to receive different EVs fractions according to the results of ultracentrifugation. To clarify this approach and achieved results corrections were added to the Conclusions section.
Besides this method was previously tested and described by our group demonstrating its effectiveness in elimination of residual cells. Clarifications are added in the Materials and Methods section.
Corrections:
Materials and methods section. All the control samples, detergent effectiveness treatment, possible false-positive or false-negative controls were performed in accordance with the previously published requirements and as previously described by our group.
Conclusions section. Phenotyping and quantification of plasma EVs using centrifugation aimed to eliminate residual cells and high-sensitivity multicolor flow cytometry could be the convenient tools for COVID-19 diagnostics.
The obtained plasma samples were stained with 16 monoclonal antibodies against a number of CD markers and analyzed using a Cytoflex S flow cytometer (Beckman Coulter, USA). Phrase "multicolor flow cytometry" in the manuscript title do not have a special meaning and rather mislead the reader. Although the labels bound to 16 used monoclonal antibodies had different spectral properties, the samples were stained with antibodies only individually, and the “multicolor” function was not used in any way. In my opinion, the word "multicolor" should be removed from the title.
Despite the using of multicolor approach of staining EVs we did not use it in the analysis. Thus, we are agree with this remark. The title was corrected.
New title:
The significance of phenotyping and quantification of plasma extracellular vesicles levels using high-sensitivity flow cytometry during COVID-19 treatment
The choice of a set of CD markers in the study is poorly explained. The authors write the following: «We used EVs markers that are potentially associated with SARS-CoV-2 dissemination via vesicles and cell-origination markers characterizing objects from different cell types that could influence on clinical manifestation of the COVID-19». In my opinion, clarification is needed which CDs from the selected set mark the EV of which types of cells (if so), and how exactly they may be associated with the pathogenesis of COVID-19. If it is difficult to make these clarifications, the choice must be explained in some other way. In the current revision of the manuscript, the 16 CD markers appear to be randomly selected and listed in completely random order.
Because of the limitations in manuscript volume caused by “short communication” article type we had no opportunity to describe our choice of EVs and cells markers in “Introduction” section. This led us to the decision to substantiate chosen markers, achieved results and already known information about investigated EVs simultaneously in “Discussion” section.
Briefly, we suggest that all types of EVs potentially participating in COVID-19 progression could be divided in two groups. First, related with several glycoproteins and/or receptors that are essential for complete fusion event, and the second, that are able to take part in pathogenesis process particularly participating in immune response, endothelial damage, ARDS, microcirculatory disorders or cytokine storming.
CD63+, CD9+, and CD147+EVs are the representatives of the first group of markers, while erythrocyte derived (CD235a+), platelets derived (CD41+), leucocytes subsets derived (CD45+, CD3+, CD4+, CD8+, CD19+, CD14+), and endothelial associated (CD146+, CD31+, CD34+, CD90, CD105) EVs are the members of the second group. Actually results of plasma EVs phenotyping is limited because of the appearance of a new generation high-sensitivity flow cytometers that are able to detect immunostained EVs over 100-200 nm in diameter only for the last few years. Besides traditionally most of the investigations used another approach of EVs isolation – ultracentrifugation. This method in this application has some limitations, for instance, need for ultracentrifuge with highly qualified personnel and additional time needed for isolation. Besides ultracentrifugation could lead to changed morphology of the EVs as well as to necessity of analysis of different fractions of the objects (microvesicles and exosomes). Taking together these explain the limited publication number of investigation of EVs in different pathological conditions and clinical applications. Thus, we cited the manuscripts proving possible participation of erythrocyte-, platelet-, leucocyte-derived and endothelial associated EVs in such pathological conditions and syndromes associated with COVID-19 as systemic inflammation, coagulation disorders, endothelial dysfunction etc.
To clarify our choice of used markers and to improve the comprehension of the manuscript we performed some changes in structure of the Discussion section.
Fig. 1 gives results on the number of vesicles positive to various CDs, and compares the groups "healthy donors", "moderate COVID-19" and "severe COVID-19". The histograms are shown only for 13 markers out of 16. At the same time, CD3, CD4, CD19, CD90 and CD105 - positive vesicles are registered with a frequency of the order of 1 event per 1 μl of the sample (i.e., essentially not registered), and do not show differences between the groups. In my opinion, these results should not be included in Fig. 1, as they have no meaning.
Thank you for the remark. The reason for the presentation of these results was the absence of the information about concentrations of immunostained plasma extracellular vesicles in donors or COVID-19 patients. Moreover, results describing influence of plasma extracellular vesicles on pathogenesis of pathological conditions and syndromes also very limited. These lead us to the decision to include this data in the manuscript, despite the insignificant changes, and hoping that it could be useful for investigators even describing healthy donors.
Figure 1 is entitled «Concentration of extracellular vesicles…. in the debut of the infection». It means that blood samples were taken at the beginning of hospital treatment of patients. However, the moment of hospitalization can in no way be called the debut of infection. Obviously, it was necessarily preceded by a period of home treatment of certain duration. Therefore, it is better to change the name of the picture to "…start of treatment….". And in the characteristics of patients in "Materials and Methods", if possible, indicate information about how much time has passed since infection (or at least since a positive PCR test).
The title of the figure 1 was changed to “Concentration of plasma extracellular vesicles (EVs) in healthy donors, patients with moderate or severe COVID-19 in the beginning of in-patient department treatment, Events/µL“. Besides we made correction clarifying that first time point was set before in-patient department treatment. These corrections were performed everywhere in the text.
Figures 2 and 3 are exactly the same type and are called the same - «Dynamics in plasma concentration of extracellular vesicles (EVs) in COVID-19 patients during the treatment comparing with healthy donors (HD)». Only CD markers differ - there are 3 of them in Fig. 2 (CD63 +, CD9 +, CD147 +) and 11 others in Fig. 3. It is not clear what the authors were guided by when dividing this data array into 2 figures, and why they divided the data in this way.
Figures 2 and 3 were combined.
In addition, I was unable to find information in the manuscript whether these results on the dynamics of CD markers in extracellular vesicles relate to a severe or moderate course of COVID-19, though this is very important.
Understanding the importance of the point we have launched the new project with increased number of patients where we will focus on searching new markers demonstrating COVID-19 severity.
Further, the total duration of observation in this case was 12-18 days. It is quite obvious that during this time the condition of some patients should have improved (according to the criteria chosen by the authors), and they should have moved from the “severe course” group to the “moderate course” group, and the condition of some patients possibly should have worsened, with the opposite dynamics. In graphs shown in Fig. 2 and 3, these most important factors are not indicated or commented on.
The decision about belonging of each patient to a particular group was made while admitting in the in-patient department. All the patients had positive clinical dynamics and were safely discharged from the hospital. Appropriate corrections were added to the “Materials and methods” section.
Corrections
All the patients received oxygen therapy depended on the patient’s condition and ranged from 1 till 21 days. 49 % patients received short course of prednisolone or dexamethasone therapy for up to 4 days. During the treatment, all the patients from both groups had positive clinical dynamics and were safely discharged from the hospital.
Regarding to Fig. 3, the authors write “The treatment-related dynamics of only CD235a+, CD4+, CD8+, CD19+, and CD146+ EVs were significant (Figure 3) but among them just CD235a+ and CD19+ are able to be reasonably detectable». If so, then serious questions arise about the significance of all the data in the article in Fig. 1, 2, 3, except for those related to two well-detectable proteins. To clarify the performance and specificity of the antibodies used and to further verify the research results, it would be advisable to demonstrate positive controls and use an alternative research method, for example, Western blotting.
Unfortunately the information about concentrations and other parameters of immunostained plasma EV detected using method of high-sensitivity flow cytometry is very limited in the case of healthy donors and absolutely absence in the case of COVID-19 patients. These lead as to conclusion to demonstrate maximum achieved results about concentrations of EV in plasma of healthy donors and in COVID-19 patients.
While performing the experiments all the control samples, possible false-positive or false-negative controls were performed in accordance the previously published requirements (1, 2). All these procedures were described in details in our previous publications (3, 4). Unfortunately “Short Communication” type of the manuscript is not allowed to describe all used methods in details. However we are agree that some additional methods are necessary to clarify the results. Thus, we prepared supplementary figure 1 demonstrating EV lost after detergent Triton X100 treatment. This test confirms presence of membranes in detected events.
References
- Welsh JA, Van Der Pol E, Arkesteijn GJA, Bremer M, Brisson A, Coumans F, et al. MIFlowCyt-EV: a framework for standardized reporting of extracellular vesicle flow cytometry experiments. J Extracell Vesicles [Internet]. 2020;9(1). Available from: https://doi.org/10.1080/20013078.2020.1713526
- Théry C, Witwer KW, Aikawa E, Alcaraz MJ, Anderson JD, Andriantsitohaina R, et al. Minimal information for studies of extracellular vesicles 2018 (MISEV2018): a position statement of the International Society for Extracellular Vesicles and update of the MISEV2014 guidelines. J Extracell Vesicles. 2018;7(1).
- Fedorov A, Kondratov K, Kishenko V, Mikhailovskii V, Kudryavtsev I, Belyakova M, et al. Application of high-sensitivity flow cytometry in combination with low-voltage scanning electron microscopy for characterization of nanosized objects during platelet concentrate storage. Platelets [Internet]. 2020 Feb 17;31(2):226–35. Available from: https://www.tandfonline.com/doi/full/10.1080/09537104.2019.1599337
- Kondratov K, Nikitin Y, Fedorov A, Kostareva A, Mikhailovskii V, Isakov D, et al. Heterogeneity of the nucleic acid repertoire of plasma extracellular vesicles demonstrated using high-sensitivity fluorescence-activated sorting. J Extracell Vesicles [Internet]. 2020 Jan 1;9(1):1743139. Available from: https://doi.org/10.1080/20013078.2020.1743139
Corrections
All the control samples, detergent effectiveness treatment, possible false-positive or false-negative controls were performed in accordance with the previously published requirements[7][11] and as previously described by our group [10][12]. Stained samples dilution was performed in accordance of prevention of coincidence effects as previously described[10][13][12]. Detergent sensitivity of EVs is presented in the supplementary figure 1
The “Discussion” section is well structered, it demonstrates knowledge and appropriate citation of literature on the problem, but practically does not focus on the authors' own data. It is necessary to analyze and logically link the obtained own results with the existing level of knowledge on the problem, and not be limited to a simple statement of certain changes in the concentrations of CD - markers on the surface of extracellular vesicles.
We came to the conclusion to prepare the manuscript as “short communication” taking into account the high social significance of all new knowledge about COVID-19 pandemia and SARS-CoV-2 as well as very limited information about participation of plasma extracellular vesicles in pathogenesis of viral infection or clinical manifestation. We suppose that despite the limited number of some used laboratory methods the presented results could be valuable for investigators of COVID-19 and presented method of phenotyping and quantification of plasma EV could be useful for clinical application.
In lane 158 it is written that “Results of Gunasekaran demonstrated that COVID-19 virus infected cells produce exosomes containing virus particles [21]”. I didn’t find any information in the reference [21] about SARS-CoV-2 virus particles in exosomes. Moreover the manuscript was published online on 2020 Jan 21 and can’t contain any experimental data on SARS-CoV-2. The further statements on EVs, that contain virus particles, are also controversial. EVs from the virus-infected cells are able to contain virus proteins and RNA fragments, some (+)RNA viruses use intracellular EVs maturation pathways for replication and morphogenesis.
We apologize for the inaccuracy. These were the patients not with SARS-CoV-2 but with coronavirus. Corrections were added in the manuscript.
We did not investigate the presence of viruses or virus RNA in the extracellular vesicles of our patients but we discuss this possibility according to the results of previously published manuscripts.
Correction
Gunasekaran et al. demonstrated that circulating exosomes isolated from patients with respiratory viral infection contained specific viral antigens of respiratory syncytial virus, coronavirus and rhinovirus respectively[1].
In my opinion, the manuscript contains poorly systematized and insufficiently analyzed data and it is necessary to exclude the unsignificant results. The manuscript requires major revision.
To improve the comprehension of the manuscript and increase the interest to performed results we made corrections and additions in the text especially in the “discussion” section. We hope that these corrections will be useful and, at the same time, the volume of the manuscript will not exceed “short communication” type.

Reviewer 2 Report
In this manuscript Golovkin and co-authors present an analysis of extracellular vesicles (EV) contained in plasma of COVID-19 patients. They use a modern cytometer to analyze the presence of many surface markers at different stages of the disease and during the treatment with steroids. They underline differences in the number of vesicles carrying typical EV, red blood cell and lymphoid markers. They also tried to determine the size of the particles present in their samples and speculate that they may contain and spread viral particles.
I see some points that need to be discussed:
the number of vesicles counted in figure nr.1 and nr.2, 3 appears different. In figure nr.1 the legend says events/ul, in nr.2 and nr.3 events/ml. Can the authors comment on that?
Line 158-159 Is this statement really supported by convincing data? In the cited paper extracellular vesicles are apparently isolated by classical methods that do not discriminate between viruses and small vesicles. Line 163-164 in the cited papers Coronavirus containing vesicles are not described as being released and circulate. They look more like virus-induced alterations in cells. In different parts of the paper the authors suggest that viral particles may be present in small and medium sized extracellular vesicles but no clear bibliography is provided and no viral markers are used in the experiments. Actually the detection of CoV-2 determinats would have been relatively easy to achieve.
without an appropriate characterization of vesicles I believe that they can't be presented as vehicle for virus spreading, but just as possible dignostic markers.
Author Response
We appreciate reviewer for the comments and questions. We are pleased that our manuscript have triggered so many questions. We hope that our answers will be useful and all corrections will increase the quality of the manuscript.
Prof. Alexey Golovkin and co-authors.
In this manuscript Golovkin and co-authors present an analysis of extracellular vesicles (EV) contained in plasma of COVID-19 patients. They use a modern cytometer to analyze the presence of many surface markers at different stages of the disease and during the treatment with steroids. They underline differences in the number of vesicles carrying typical EV, red blood cell and lymphoid markers. They also tried to determine the size of the particles present in their samples and speculate that they may contain and spread viral particles.
I see some points that need to be discussed:
the number of vesicles counted in figure nr.1 and nr.2, 3 appears different. In figure nr.1 the legend says events/ul, in nr.2 and nr.3 events/ml. Can the authors comment on that?
Thank you for your remark. The correct vesicle count is events/µL. Corrections were made in the appropriate figure.
Line 158-159 Is this statement really supported by convincing data? In the cited paper extracellular vesicles are apparently isolated by classical methods that do not discriminate between viruses and small vesicles.
In this section we tried to explain our choice of used EVs markers. On the one hand tetraspanins are well-known markers of EVs (1) and on the other hand proteins could be incorporated into EVs also with participation of tetraspanins. Thus, according to cited publications, the presence of virus proteins in an EVs is suggested.
- Théry C, Witwer KW, Aikawa E, Alcaraz MJ, Anderson JD, Andriantsitohaina R, et al. Minimal information for studies of extracellular vesicles 2018 (MISEV2018): a position statement of the International Society for Extracellular Vesicles and update of the MISEV2014 guidelines. J Extracell Vesicles. 2018;7(1).
Line 163-164 in the cited papers Coronavirus containing vesicles are not described as being released and circulate. They look more like virus-induced alterations in cells. In different parts of the paper the authors suggest that viral particles may be present in small and medium sized extracellular vesicles but no clear bibliography is provided and no viral markers are used in the experiments. Actually the detection of CoV-2 determinats would have been relatively easy to achieve.
We investigated the extracellular vesicles of different cell origin during COVID-19 assuming their participation in infection pathogenesis. Meanwhile some authors suggest that plasma membrane particles could be related to viruses and virus infection progression. While discussing our results we found it necessary to mention this possible explanation.
without an appropriate characterization of vesicles I believe that they can't be presented as vehicle for virus spreading, but just as possible diagnostic markers.
We agree that statement about possible usage of EV as a vehicle for viruses could be declared only after additional investigations. We suggest that phenotyping and quantification of plasma EVs could be useful method for COVID-19 diagnostics and monitoring.